# A Stable Rechargeable Aqueous Zn–Air Battery Enabled by Heterogeneous MoS_2_ Cathode Catalysts

**DOI:** 10.3390/nano12224069

**Published:** 2022-11-18

**Authors:** Min Wang, Xiaoxiao Huang, Zhiqian Yu, Pei Zhang, Chunyang Zhai, Hucheng Song, Jun Xu, Kunji Chen

**Affiliations:** 1National Laboratory of Solid State Microstructures, School of Electronics Science and Engineering, College of Engineering and Applied Sciences, Nanjing University, Nanjing 210093, China; 2School of Materials Science and Chemical Engineering, Ningbo University, Ningbo 315211, China; 3College of Electrical and Information Engineering, Zhengzhou University of Light Industry, Zhengzhou 450002, China

**Keywords:** Zn–air batteries, 2D MoS_2_, defects-embedded, OER, ORR

## Abstract

Aqueous rechargeable zinc (Zn)–air batteries have recently attracted extensive research interest due to their low cost, environmental benignity, safety, and high energy density. However, the sluggish kinetics of oxygen (O_2_) evolution reaction (OER) and the oxygen reduction reaction (ORR) of cathode catalysts in the batteries result in the high over-potential that impedes the practical application of Zn–air batteries. Here, we report a stable rechargeable aqueous Zn–air battery by use of a heterogeneous two-dimensional molybdenum sulfide (2D MoS_2_) cathode catalyst that consists of a heterogeneous interface and defects-embedded active edge sites. Compared to commercial Pt/C-RuO_2_, the low cost MoS_2_ cathode catalyst shows decent oxygen evolution and acceptable oxygen reduction catalytic activity. The assembled aqueous Zn–air battery using hybrid MoS_2_ catalysts demonstrates a specific capacity of 330 mAh g^−1^ and a durability of 500 cycles (~180 h) at 0.5 mA cm^−2^. In particular, the hybrid MoS_2_ catalysts outperform commercial Pt/C in the practically meaningful high-current region (>5 mA cm^−2^). This work paves the way for research on improving the performance of aqueous Zn–air batteries by constructing their own heterogeneous surfaces or interfaces instead of constructing bifunctional catalysts by compounding other materials.

## 1. Introduction

Zinc–air (Zn–air) batteries using oxygen (O_2_) as the active medium have recently attracted extensive research interest as a promising energy storage device for the next-generation energy storage technology due to their high theoretical energy density of 1086 Wh kg^−1^, safety and environmental friendliness, cost-effectiveness, and readily available raw materials [1,2,3,4,5,6]. The reported Zn–air battery usually consists of a Zn metal anode (Zn-anode), an alkaline aqueous electrolyte, and a porous carbonaceous cathode, where the O_2_ can be absorbed and react with the electron and H_2_O to convert OH^−1^ via an oxygen reduction reaction (ORR, see Equation (1)) on discharge, showing a theoretical specific energy of ~1086 Wh kg^−1^, being at least four times higher than that of Li-ion batteries of ~265 Wh kg^−1^ [7,8,9].
ORR on discharge: O_2_ (g) + 2H_2_O (l) + 4e^−1^ → 4OH^−^ (aq)(1)
OER on charge: 4OH (aq) → O_2_ (g) + 2H_2_O (l) + 4e^−^
(2)

On charge, the OH^−1^ can be converted into O_2_ gas and H_2_O to store electric energy via the oxygen evolution reaction (OER, see Equation (2)) [8]. However, the OER and ORR of the cathodes are slow in kinetics due to the proton-coupled multistep electron transfer process for the reversible charge/discharge reactions [7,8,9,10,11,12,13]. In addition, the alkaline aqueous electrolyte can capture carbon dioxide in the air and generate insoluble and insulating carbonate [14], such as the typical Li_2_CO_3_ byproduct that has a ~5.09 eV band gap defined by the highest occupied molecular orbital (HOMO) and the lowest unoccupied molecular orbital (LUMO) levels [15], which result in extremely sluggish kinetics for OER and ORR processes [14]. Various advanced strategies have been used to improve the kinetics of the OER/ORR on charge/discharge [16]. Among them, precious metal and metal oxide catalysts, including platinum (Pt), ruthenium (Ru), iridium (Ir), ruthenium dioxide (RuO_2_), iridium dioxide (IrO_2_), etc., usually show good catalytic activity for ORR or OER. As the most representative cooperation, Pt is used as an ORR electrocatalyst in alkaline media, and the ruthenium dioxide (RuO_2_)/iridium dioxide (IrO_2_) is used as the OER catalyst [17,18,19,20,21]. However, these precious metals and metal oxides are difficult to use in large-scale industrial applications due to their poor chemical stability, relative scarcity, and high cost [21]. Therefore, the exploitation of cheap and efficient electrocatalysts to boost the OER and ORR processes is highly necessary for developing high-performance aqueous Zn–air batteries.

As one of the most classical 2D transition-metal chalcogenides (TMDs), molybdenum disulfide (MoS_2_) has abundant active edge sites and good crystallinity [22,23], and it recently has been considered to be an effective hydrogen evolution reaction (HER) catalyst [24]. In addition, research suggests that crystal-amorphous interface and grain boundaries of MoS_2_ can catalyze electrochemical reactions [22]. Nonetheless, Amiinu et al. recently reported multifunctional Mo–N/C@MoS_2_ electrocatalysts for HER, OER, ORR, and Zn–air batteries that show a high power density of ≈196.4 mW cm^−2^ and a voltaic efficiency of ≈63% at 5 mA cm^−2^, as well as excellent cycling stability, even after 48 h at 25 mA cm^−2^ [10]. Bai et al. reported heterostructure Co_9_S_8_@MoS_2_ core–shell structures that exhibited robust OER performance and a 20 h cycle lifespan for Zn−air batteries with low high discharge voltages/low discharge voltages (~1.28 V/2.03 V) [25]. Plulia et al. reported layered MoS_2_/graphene nanosheet catalysts that showed enhanced oxygen reduction activity, nearly double that of graphene nanosheets or ∼25-fold that of MoS_2_ nanosheets, and high open circuit voltages (1.4 V) and high specific energy of up to 130 W h kg^−1^ for assembled Zn−air batteries [26]. Although great efforts have thus been focused on improving the electrocatalytic performance of MoS_2_, the OER activity of MoS_2_ is largely limited, which makes it difficult to use for OER and Zn–air batteries [27]. Compared with the reported bifunctional catalysts consisting of two typical catalytic function materials, the development of a single and efficient non-precious metal catalyst with a good oxygen evolution reaction and oxygen reduction reaction, by a simple preparation process, is of great significance for the development and application of aqueous zinc–air batteries.

Here, we report an aqueous Zn–air battery that shows advantages in cycle performance and economic cost. The Zn–air battery consists of a Zn-anode, an alkaline aqueous solution electrolyte, and the air cathode (see Appendix A). The air cathode materials, consisting of electron-conducting carbon nanotube (CNT) and MoS_2_ catalysts, are prepared on the carbon paper gas diffusion layer (GDL) that can provide an efficient gas transport for the assembled battery (see Figure 1a). A stable cycling Zn–air battery can be obtained through several mechanisms: (a) An air cathode consisting of heterogeneous MoS_2_ catalysts and CNT on the GDL can provide good conductivity for charge transport, and an adequate three-phase interface for ORR and OER for the assembled aqueous Zn–air battery on charge/discharge (see Figure 1b) [21]. (b) A heterogeneous interface consisting of a super-hydrophobic carbon paper GDL and a hydrophilic MoS_2_ catalyst enables efficient gas utilization, where the oxygen (O_2_) can be directly absorbed and reacted at the interface without being dissolved in aqueous solution on the charge/discharge process (see Figure 1c,d) [28]. (c) The MoS_2_ with heterogeneous and abundant active edge sites can enable an efficient electrochemical reaction, especially for the hydrogen evolution reaction (HER) and the oxygen evolution reaction [29]. Notably, the heterogeneous catalyst is only composed of MoS_2_ that can form heterogeneous interfaces and defects-embedded active edge sites only by using a relatively low synthesis temperature (see Section 2). The oxygen evolution activity of the heterogeneous MoS_2_ catalyst, by constructing its own heterogeneous surface or interface instead of constructing bifunctional catalysts by compounding another OER catalytically active material, exceeds commercial Pt/C-RuO_2_ catalytic activity for the assembled aqueous Zn–air battery.

## 2. Materials and Methods

### 2.1. Chemicals

(NH_4_)_2_MoS_4_, dimethylformamide (DMF), ethanol, carbon nanotubes, RuO_2_, and KOH were purchased from Sigma Aldrich. Nafion D520 (5 wt%), and carbon paper (GDL340) were received from SCI-Materials-Hub. A commercial Pt/C catalyst with 20 wt% Pt was purchased from Shanghai Macklin Biochemical Technology Co., Ltd (Shanghai, China). All the reagents were of analytical grade and used as received without further purification. Deionized water was used throughout the experimental processes.

### 2.2. Material Preparation

MoS_2_ was prepared by hydrothermal synthesis. Briefly, 0.2 g (NH_4_)_2_MoS_4_ was dissolved in 15 mL DMF by ultrasonication for 15 min to form a homogeneous solution. Then, the solution was transferred into a 25 mL Teflon-lined stainless steel autoclave maintained at 180 °C for 10 h and cooled to room temperature naturally. The final product was collected by centrifugation, washed with water and ethanol three times each, and subsequently dried in a vacuum oven at 60 °C overnight, resulting in black powder.

### 2.3. Material Characterization

The crystal structures of samples were identified on a BRUKER D8 ADVANCE X-ray diffractometer with Cu-Kα radiation (λ = 1.54178 Å) over the 2θ range from 5 to 90°. The morphology and microstructure were characterized by a scanning electron microscope (SEM, Hitachi SU8010) and a transmission electron microscope (TEM, TecnaiG2F20).

### 2.4. Electrode Preparation and Battery Assembly

Firstly, 0.5 mL Nafion D520 solution, 1 mg MOS_2_, 1 mg carbon nanotubes, and 10 mL ethanol were measured into a beaker and ultrasonicated for 30 min. After the mixture was evenly mixed, 1 mL of the mixture was taken and sprayed evenly onto 2 × 2 cm carbon paper with a spray gun. While spraying, the surface of the carbon paper was dried with a baking lamp. After drying at 60 °C, the loading capacity of MoS_2_ catalyst was estimated to be 1 mg cm^−2^, which is the air electrode. When making the contrast electrode, it was only necessary to change the MoS_2_ catalyst with the same amount of Pt/C-RuO_2_ powder (m(Pt/C):m(RuO_2_) = 1:1), and the other steps were exactly the same as above. A polished zinc foil (thickness: 0.25 mm) was used as the anode, and the electrolyte was 6.0 M KOH for the primary Zn–air battery and 6.0 M KOH with 0.20 M Zn (CH_3_COO)_2_ for the rechargeable Zn–air battery. Measurements were carried out at 25 ℃ with an electrochemical workstation (CHI 660E, CH Instrument, Austin, TX, USA).

### 2.5. Electrochemical Test

The discharge/charge cycling performance of the ASS Zn–air battery was obtained by using an electrochemical test system (Hokuto Denko Corporation, HJ1001SD8, Meguro-ku, Tokyo). Alternating-current impedance spectroscopy of the lithium–air batteries was investigated using an electrochemical workstation (CHI 660E, CH Instrument, Austin, TX, USA).

## 3. Results

Typically, heterogeneous MoS_2_ catalysts are prepared by a simple hydrothermal process (see Appendix A). The structural and morphological properties of the prepared MoS_2_ catalysts were characterized by a transmission electron microscope (TEM), X-ray diffraction (XRD), and an aberration-corrected high-angle annular dark-field scanning TEM (HAADF-STEM). As shown in Figure 2, the prepared MoS_2_ catalysts were typical layered structures (see Figure 2a) [30] consisting of Mo and S elements, where the S and Mo elements were evenly distributed on the prepared MoS_2_ catalyst (see Figure 2a–d). XRD patterns further suggested the fact that the prepared catalysts were MoS_2_, where XRD diffraction peaks located at 14.1°, 33.2°, 39.4° and 58.2° corresponded to the (002), (100), (103), and (110) planes of the 2H-MoS_2_ nanosheet (Powder Diffraction File no. 37-1492, Joint Committee on Powder Diffraction Standards), respectively (see Figure 2e) [31]. Other diffraction peaks located at 11.4°, 22.5°, and 29.6° could be attributed to the 2m-1T phase transition of MoS_2_ driven by in situ intercalation of ammonium or alkyl amine cations [32]. High-resolution HAADF-STEM images further indicated that the prepared MoS_2_ were heterogeneous, where amorphous phases were distributed on the crystalline MoS_2_ phases (see Figure 2f) [33]. Further, the STEM of the MoS_2_ catalyst showed a heterogeneous surface consisting of active edge sites and abundant defects/disordered phases (see Figure 2g) that could enhance the electrochemical reaction, where the interplanar lattice spacings of 0.678 nm (see Figure 2g,h) were consistent with the crystal reflection of MoS_2_ [34]. This result was also consistent with the XRD result of the prepared MoS_2_ catalyst. The high-resolution HAADF-STEM and TEM also indicated the formation of heterogeneous crystalline-amorphous MoS_2_ (see Figure 2i,j). The above results suggest that the formation of heterogeneous MoS_2_ consists of crystalline-amorphous interfaces and defects-embedded active edge sites [35].

Befitting the electrochemical performance of the prepared MoS_2_, the aqueous Zn–air battery using the MoS_2_ catalyst showed a lower discharge potential (~1.17 V) and a higher charge potential (~2.39 V) than the Zn–air battery using commercial Pt/C-RuO_2_ catalysts (~1.35 V and 1.89 V, see Figure 3a) at the first cycle. Notably, the Zn–air battery using the MoS_2_ catalyst showed a lower discharge potential (~1.3 V) and a slightly lower charge potential (~1.88 V) than the Zn–air battery using commercial Pt/C-RuO_2_ catalysts (~1.26 V and 1.94 V, see Figure 3a) after 500 cycles. This indicates that the heterogeneous MoS_2_ catalysts can enable a more stable and better electrochemical performance than the expensive Pt/C-RuO_2_ catalysts for the assembled aqueous Zn–air battery [36]. In addition, the Zn–air batteries using the MoS_2_ catalysts could stably discharge a specific capacity of ~330 mAh g^−1^, ~660 mAh g ^−1^, ~3300 mAh g^−1^, and ~6600 mAh g^−1^ at 0.5 mA cm^−2^, 1 mA cm^−2^, 5 mA cm^−2^, and 10 mA cm^−2^, respectively (see Figure 3b). This suggests that the Zn–air battery using MoS_2_ can be stably operated at high current density. Note that by using heterogeneous MoS_2_ catalysts, the assembled aqueous Zn–air battery demonstrated good cycle performance with lower potential than that of the Zn–air battery using commercial Pt/C-RuO_2_ catalysts (see Figure 3c and Appendix A). The long-term electrochemical test of the Zn–air battery using the MoS_2_ catalyst showed an increasingly enhanced electrochemistry performance where the discharge potential decreased from ~2.38 V at the first cycle to ~1.98 V at the 500th cycle with a limited specific capacity of ~330 mAh g^−1^ (see Figure 3d). The enhanced electrochemical performance could also be observed by the reduction of charge potential during cycling (see Figure 3e). This indicates the fact that the MoS_2_ catalysts with heterogeneous interfaces and defects-embedded active edge sites can demonstrate better OER performance for assembled Zn–air batteries than expensive Pt/C-RuO_2_ catalysts during long-term limited capacity cycling [37]. Such good cycle performance can be attributed to the heterogeneous interface and defects-embedded active edge sites of the prepared MoS_2_ catalyst, where the edge sites of the MoS_2_ nanosheets can enable stronger adsorption toward oxygen (O_2_), and other intermediates [38,39], defects [40], and amorphous phases can make the MoS_2_ catalyst maintain high electrochemical activity [41], thus resulting in excellent cycle performance for the assembled aqueous Zn–air batteries.

Further, the heterogeneous MoS_2_ were employed as cathode catalysts of a typical aqueous Zn-O_2_ battery and a Zn-CO_2_ battery. As showed in Figure 4a, all the batteries using the MoS_2_ catalysts could operate at 0.5 mA cm^−2^ with a limited capacity of ~330 mAh g^−1^. Notably, the aqueous Zn-air battery showed the highest discharge potential (~1.17 V) and the lowest charge potential (~2.39 V) at the first cycle compared to the Zn-O_2_ battery (with a 1.3 V discharge potential and 2.2 V charge potential, see Figure 4a) and the Zn-CO_2_ battery (with a 1.18 V discharge potential and 2.16 V charge potential, see Figure 4a). Additionally, the Zn-air battery using the heterogeneous MoS_2_ catalysts demonstrated a long cycle lifespan (500 cycles, see Figure 4b), being at least four times more than the Zn-O_2_ battery (~100 cycles, see Appendix A) and Zn-CO_2_ battery (~100 cycles, see Appendix A). Clearly, the Zn-air battery using the heterogeneous MoS_2_ catalyst showed the best cycle performance and the lowest potential gap of ~0.8 V compared to the Zn-O_2_ battery and the Zn-CO_2_ battery of more than 1.0 V (see Figure 4c and Appendix A). In addition, the aqueous Zn-air battery showed a lower charge potential (~1.35 V) and a higher discharge potential (~1.89 V) than the Zn-air battery using commercial Pt/C-RuO_2_ catalysts (~1.35 V and 1.89 V, see Figure 3a) at the first cycle. Notably, the Zn-air battery using the MoS_2_ catalyst showed a lower discharge potential (~1.3 V) and a slightly lower charge potential (~1.88 V) than the Zn-air battery using commercial Pt/C-RuO_2_ catalysts (~1.26 V and 1.94 V, see Figure 3a) after 500 cycles. Furthermore, the assembled Zn-air battery showed a long cycle lifespan of over 500 cycles, being at least five times higher than that of the assembled Zn-O_2_ battery and the Zn-CO_2_ battery (~100 cycles, see Figure 4d). Notably, the Zn-air battery showed enhanced electrochemical cycle performance during long-term cycling that could be attributed to the activation of the cathode materials [42]. This indicates that the heterogeneous MoS_2_ can be successfully used as cathode catalysts for both the Zn-O_2_ battery and the Zn-CO_2_ battery.

## 4. Conclusions

In summary, by simple hydrothermal synthesis, we prepared hydrophilic and heterogeneous MoS_2_ catalysts consisting of crystalline-amorphous interfaces and defects-embedded active edge sites that enables a good three phase interface on carbon paper GDL, and efficient O_2_ and CO_2_ utilization by their hydrophilic characteristics (with a 24 degree water contact angle) and the superhydrophobic characteristics of carbon paper GDL (with a 130 degree water contact angle). Such MoS_2_ catalysts showed decent oxygen evolution and acceptable oxygen reduction catalytic activity compared to commercial Pt/C and RuO_2_, which enabled a cycling durability of 500 cycles (~180 h) for an assembled aqueous Zn-air battery at 0.5 mA cm^−2^ with a limited capacity of 330 mAh g^−1^, and lower charge potentials (~1.88 V after 500 cycles) than the Zn-air battery using expensive Pt/C and RuO_2_ after cycles. Notably, the Zn-air battery using the prepared MoS_2_ catalysts could operate stably even at a large current density of 10 mA cm^−2^. The Zn–Air battery using single MoS_2_ catalyst also shows comparable performance among the Zn–Air batteries using MoS_2_-based catalysts (see Appendix A). In addition, the heterogeneous MoS_2_, as an effective cathode catalyst, could catalyze the reversible circulation of the Zn-O_2_ battery and the Zn-CO_2_ battery, demonstrating that the heterogeneous MoS_2_ catalyst can potentially replace Pt/C and RuO_2_ catalysts in aqueous rechargeable Zn-air batteries, Zn-O_2_ batteries, and Zn-CO_2_ batteries.

## Figures and Tables

**Figure 1 nanomaterials-12-04069-f001:**
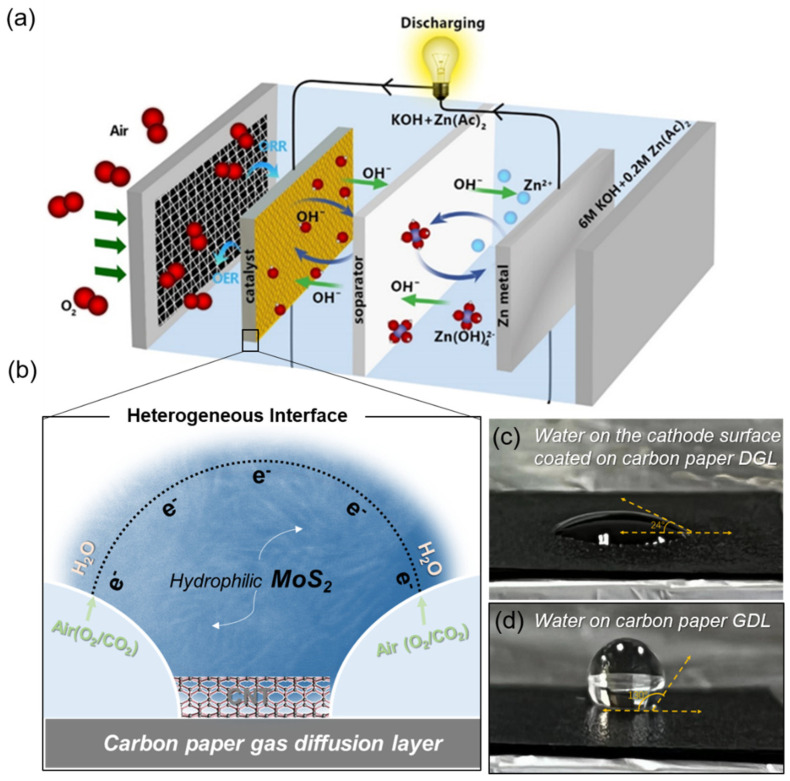
Configuration of the aqueous Zn-air battery with a heterogeneous design. (**a**) Configuration of the aqueous Zn-air battery and the enlarged diagram of the heterogeneous interface consisting of hydrophilic MoS_2_ catalysts, electron-conductive CNT, and a super-hydrophobic carbon paper gas diffusion layer. (**b**) Diagram of charging and discharging of an assembled water-zinc-air battery with a three-phase interface ORR and OER. (**c**) Optical photo of a drop of water on the cathode surface coated on carbon paper GDL that shows a contact angle of ~24 degrees. (**d**) Optical photo of a drop of water on carbon paper GDL that shows a contact angle of ~130 degrees.

**Figure 2 nanomaterials-12-04069-f002:**
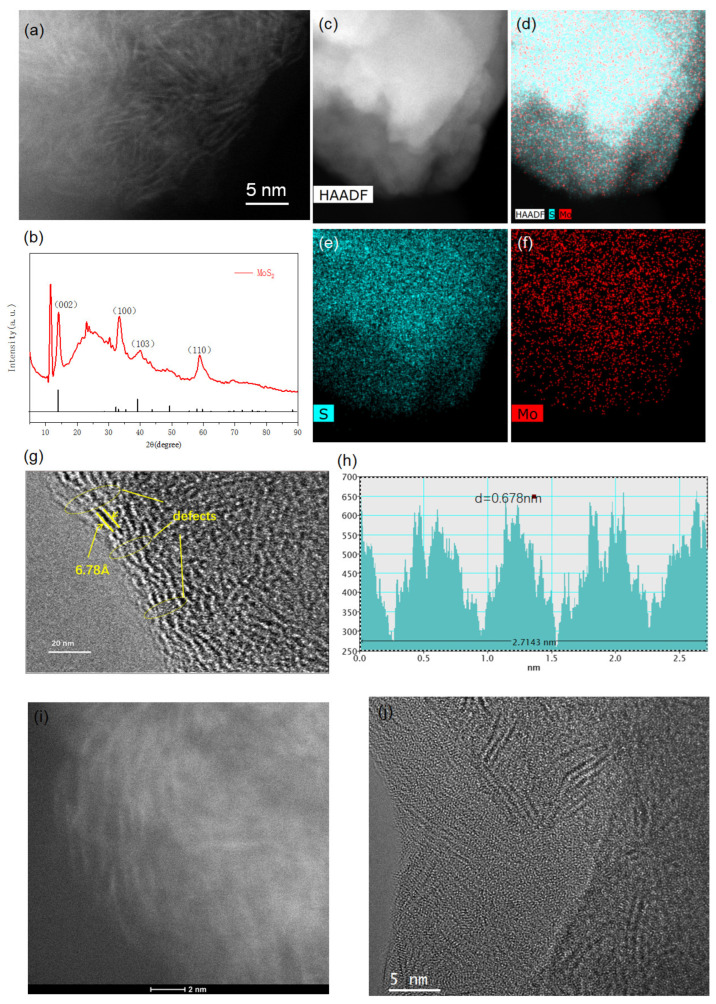
Characterization of the MoS_2_ catalyst for the Zn–air battery. (**a**) STEM of the prepared MoS_2_ catalyst. (**b**–**d**) EDS mappings of elementals (Mo and S) of the MoS_2_ catalyst. (**e**) XRD pattern of the prepared MoS_2_ catalyst. (**f**) STEM image and (**g**) TEM image of the MoS_2_ catalyst. (**h**) Interplanar spacings of the MoS_2_ catalyst, where the spacing of the nanosheets is found to be ∼6.78 Å. (**i**) High-resolution STEM and (**j**) TEM image of the prepared MoS_2_ catalysts.

**Figure 3 nanomaterials-12-04069-f003:**
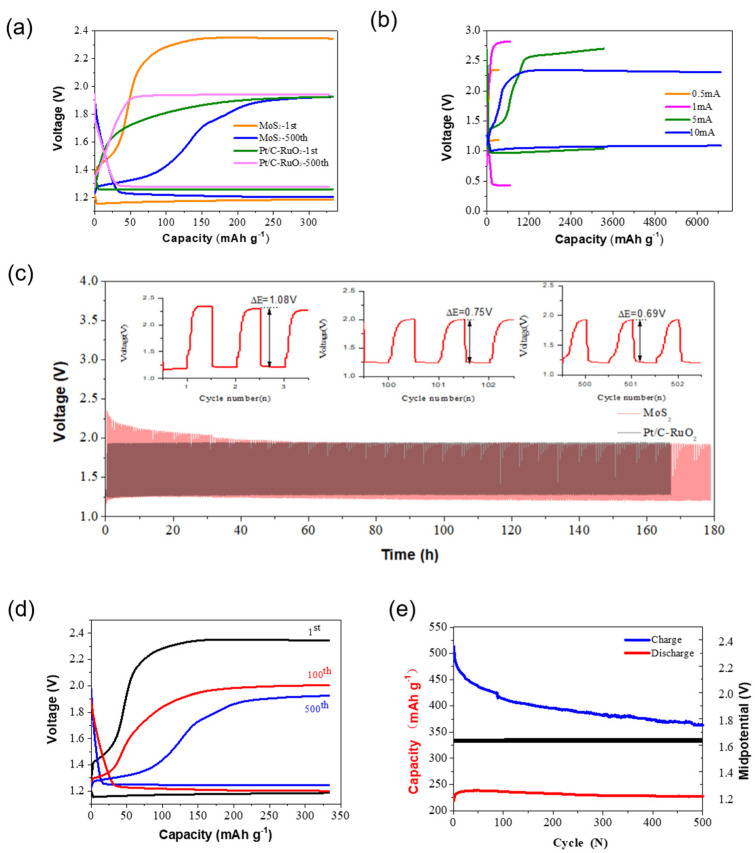
Electrochemical performance of the Zn-air batteries. (**a**) The discharge and charge curves of Zn-air batteries using the prepared MoS_2_ catalysts and commercial Pt/C-RuO_2_ catalysts at the 1st and 500th cycles, with a limited capacity of 330 mAh g^−1^, respectively. (**b**) The discharge and charge curves of the assembled Zn-air batteries using the MoS_2_ catalysts at various current densities of 0.5 A cm^−2^, 1 A cm^−2^, 5 A cm^−2^, and 10 A cm^−2^, respectively. (**c**) Electrochemical cycle performance of the Zn-air batteries using the prepared MoS_2_ catalysts and commercial Pt/C-RuO_2_ catalysts operating at 0.5 A cm^−2^, respectively. (**d**) The discharge and charge curves of Zn-air batteries using the prepared MoS_2_ catalysts at the 1st, 100th and 500th, respectively, and (**e**) corresponding to the voltage and capacity change of the Zn-air battery during the cycle.

**Figure 4 nanomaterials-12-04069-f004:**
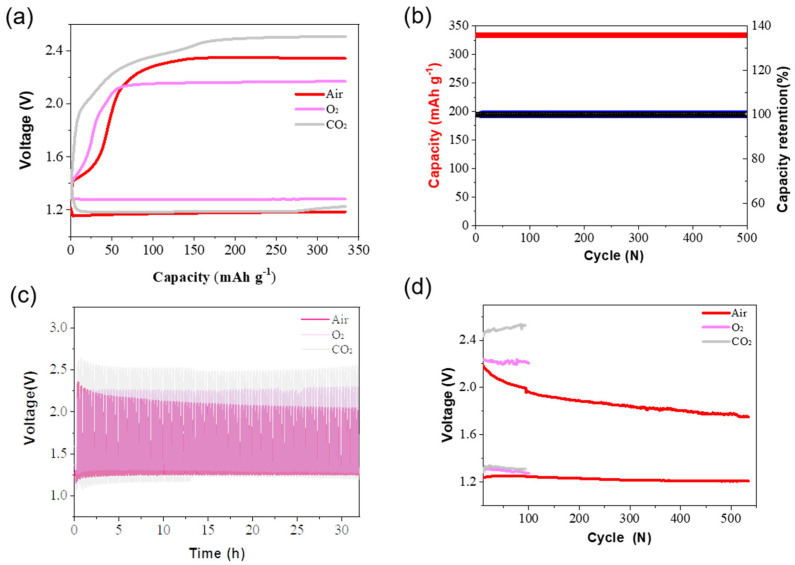
Electrochemical performance of the Zn-air battery, Zn-O_2_ battery, and Zn-CO_2_ battery using a heterogeneous MoS_2_ catalyst. (**a**) The discharge and charge curves of the Zn-air battery, Zn-O_2_ battery, and Zn-CO_2_ battery using a heterogeneous MoS_2_ catalyst at the 1st cycle with a limited capacity of 330 mAh g^−1^ and a current density of 0.5 mA cm^−2^. (**b**) The specific capacity and corresponding capacity retention of the aqueous Zn-air battery, Zn-O_2_ battery, and Zn-CO_2_ battery during cycling with a current density of 0.5 mA cm^−2^. (**c**) Voltage–time curves of the Zn–air battery, Zn-O_2_ battery, and Zn-CO_2_ battery, respectively, for cycling over 30 h with a limited capacity of ~330 mAh g^−1^ at 0.5 A cm^−2^. (**d**) The discharge and charge voltage change of the Zn-air battery, Zn-O_2_ battery, and Zn-CO_2_ battery for long-term cycling with a limited capacity of ~330 mAh g^−1^.

## Data Availability

The data presented in this study are available on request from the corresponding author.

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
