# Peer review of "A Stable Rechargeable Aqueous Zn–Air Battery Enabled by Heterogeneous MoS2 Cathode Catalysts"

_nanomaterials, 2022, doi:10.3390/nano12224069_

Round 1

Reviewer 1 Report

The article "A stable rechargeable aqueous Zn-Air battery enabled by heterogeneous MoS2 cathode catalysts" is devoted to the synthesis of highly efficient heterogeneous MoS2 catalysts for use in aqueous Zn-Air battery by a simple hydrothermal method. The structure of the obtained material has been studied in detail, the efficiency of heterogeneous MoS2 catalysts has been studied in a battery for 500 cycles, in comparison with a commercial analogue. However, a number of remarks should be noted:

Why "heterogeneous MoS2 catalysts can enable a more stable and better electrochemical performance than the expensive Pt/C-RuO2 catalysts" (line 183), if for MoS2 catalysts higher charge potential drops from 2.39 V to 1.88 V, while for commercial almost unchanged?

The text of the article contains links to Supporting Information, but they are not added to the article.

It is necessary to indicate the brand of commercial Pt/C-RuO2 catalysts. The text indicates that "Platinum on carbon (20%) was purchased from Macklin" it is necessary to explain the composition and manufacturer of the catalyst.

There are not enough literary references in the article, it is necessary to add more recent work in the field of materials for Zn-Air battery.

There are many typos in the text, e.g. Pt/ c-RuO2 (Line 118), MOS2 (Line 112, 116) missing spaces (Line 99)

Author Response

Responses to Reviewer #1:

The article "A stable rechargeable aqueous Zn-Air battery enabled by heterogeneous MoS2 cathode catalysts" is devoted to the synthesis of highly efficient heterogeneous MoS2 catalysts for use in aqueous Zn-Air battery by a simple hydrothermal method. The structure of the obtained material has been studied in detail, the efficiency of heterogeneous MoS2 catalysts has been studied in a battery for 500 cycles, in comparison with a commercial analogue. However, a number of remarks should be noted:

  1. Why "heterogeneous MoS2 catalysts can enable a more stable and better electrochemical performance than the expensive Pt/C-RuO2 catalysts" (line 183), if for MoS2 catalysts higher charge potential drops from 2.39 V to 1.88 V, while for commercial almost unchanged?

Reply:That’s a good question. We reply to your questions from the following two aspects. (1) For the more stable and better electrochemical performance of heterogeneous MoS2 catalysts consisting of crystalline/amorphous heterogeneous interface and defects-embedded active edge sites than Pt/C-RuO2, MoS2 is first a good electro-catalysts for hydrogen evolution reaction (HER), oxygen evolution reaction and metal-oxygen batteries as the previous reported research work [see Adv. Mater. 2016, 28(10), 1917−1933; Adv. Mater. Interfaces 2016, 3(9), 1500669 and Nanoscale 2018, 10 (47), 22549−22559]. Importantly, the intrinsic catalytic activity of MoS2 is determined by the adsorption ability of the edged Mo sites toward O2 species. The edge sites of the MoS2 nanosheets usually display stronger adsorption toward oxygen (O2) and other intermediates, which leads to a high electro-catalytic activity for the reversible electrochemical conversion [see ACS Nano 2016, 10, 2167−2175; Nanoscale 2018, 10, 22549−22559 and Nano Energy 2019, 65, 103996]. In addition, defect engineering and heteroatom doping are also used to improve the electrochemical activity of MoS2 to enhance the performance of metal-oxygen batteries [see Front. Energy Res. 2020, 8, 109; Nano Energy 2019, 65, 103996 and Chem. Eng. J. 2020, 382, 122854]. Notably, the presence of amorphous phase makes it easier to maintain high electrochemical activity for MoS2 catalyst. For example, Wu et al., have provided both experimental and theoretical evidence for the importance of the short Mo @ Mo bond structures of 1T and amorphous MoS2 in comparison to crystalline MoS2 for explaining the higher electrochemical performance [see the work published on ChemSusChem 2019, 12, 4383-4389]. Whereas crystalline 1T-MoS2 stabilized by intercalated Li+ also displays high performance, Li ions were found to dissolve in the electrolyte during electrochemical testing, resulting in a slow transformation back to the 2H-MoS2 phase and a concomitant decrease in HER activity. In contrast, amorphous MoS2 retains much of its high HER activity during prolonged operation [see ChemSusChem 2019, 12, 4383-4389]. As a result of these advantages mentioned above, the heterogeneous MoS2 can be used as one of the potential candidates for the O2 cathode catalyst with stable and good electrochemical performance, especially for energy storage batteries [Nano Energy 2019, 65, 103996 and Energy & Fuels 2021 35 (7), 5613-5626]. (2) The drop of the charge potential drops (from 2.39 V to 1.88 V) of the MoS2 catalysts is attributed to the electrochemical activity of the two-dimensional (2D) MoS2 catalysts with high areal surface. In our experimental preparation, the hydrothermally synthesized MoS2 nanosheet catalysts form flower-like aggregates consisting of 2D MoS2 nanosheets that usually show a large surface area. Such MoS2 catalysts with large surface area usually shows a typical activation process during discharge and charge process that result in a charge potential drop from 2.39 V to 1.88 V for MoS2 catalysts that almost unchanged charge potential for commercial Pt/C-RuO2 catalysts. Similar phenomena are often reported in nanoscale materials for energy storage batteries especially for one-dimensional and two-dimensional nanostructures [see Adv.Mater.2017, 29, 1602300 and Energy Storage Materials 38 (2021) 200–230].

  1. The text of the article contains links to Supporting Information, but they are not added to the article.

Reply:Thank you very much for your reminder. We have revised the manuscript and linked the Supporting Information to the corresponding description in the revised manuscript (see Line 82, Line 151, Line 204 and Line 231 marked yellow in the text in the revised manuscript).

  1. It is necessary to indicate the brand of commercial Pt/C-RuO2 catalysts. The text indicates that "Platinum on carbon (20%) was purchased from Macklin" it is necessary to explain the composition and manufacturer of the catalyst.

Reply:Thank you very much for your reminder. The commercial Pt/C catalyst with 20 wt% Pt was purchased from Shanghai Macklin Biochemical Technology Co., Ltd. We have also added the message in the revised manuscript (See Line 107 to Line 108 marked yellow in the text).

  1. There are not enough literary references in the article, it is necessary to add more recent work in the field of materials for Zn-Air battery.

Reply:Thanks for your reminder. According you suggestion, we have added about 10 recent research works in the field of materials for Zn-Air batteries (See Ref [4] to Ref [6], Ref [8], Ref [9], Ref [13], Ref [27], Ref [33] and Ref [35] to Ref [42] marked yellow in the References in the revised manuscript).

  1. There are many typos in the text, e.g. Pt/ c-RuO2 (Line 118), MOS2 (Line 112, 116) missing spaces (Line 99)

Reply:Thank you very much for your reminder. We have revised the typos in the revised manuscript (see Line 128, Line 130 and other places marked yellow in the text).

Reviewer 2 Report

The manuscript entitled "A stable rechargeable aqueous Zn-Air battery enabled by heterogeneous MoS2 cathode catalysts". This work is well written and presented and it has enough characterization and electrochemical data and deserves to be published in Nanomaterials. However, this work lacks some important information and discussion. Before I recommend its acceptance, some points must be clarified and a moderate revision is needed.

Some other issues that need to be addressed are:

1- The main problem statement and justification for the research has not been clearly stated.

2- It is not clear the contribution of the manuscript to the empirical literature.

3- The novelty of the work should be boldly explicated. The main novelty in this work must be clearly pointed out in the introduction.

4- The authors should mention on the concept of this work with the progress against the most recent state-of-the-art similar studies.

5- Figure 2a (STEM) is not very clear to observe. Can the authors replace it by one of better quality?  

6- Line 153: Further, the STEM of MoS2 catalyst shows a clear heterogeneous surface consisting of active edge sites and abundant defects (see Figure 2g)…..if I am not mistaken, I see more disordered and the defect highlighted in Figure2 g I am not sure is clearly seen.

7-  Line 182: “It indicates that the heterogeneous MoS2 catalysts can enable a more stable and better electrochemical performance than the expensive Pt/C-RuO2 catalysts for the assembled aqueous Zn-Air battery.”…why? A reasonable explanation should be given. 8 Some part of the text lacks explanation about assertive statements. In addition, some discussion also involving other battery systems should be compared highlighting some pros and cons. The 2 references can help the authors with that.

https://doi.org/10.3390/nano12223933

https://doi.org/10.1021/acsomega.2c06054

9-  Figure 4d is not discussed and it has to do so.

10-  The is not Supporting Information file, or at least I had no access to it.

Author Response

Responses to comments of Reviewer #2:

The manuscript entitled "A stable rechargeable aqueous Zn-Air battery enabled by heterogeneous MoS2 cathode catalysts". This work is well written and presented and it has enough characterization and electrochemical data and deserves to be published in Nanomaterials. However, this work lacks some important information and discussion. Before I recommend its acceptance, some points must be clarified and a moderate revision is needed.

Reply:Thanks for recommending our work published on Nanomaterials. According to your comments, we have provided some important information and discussion, and revised the manuscript.

Some other issues that need to be addressed are:

  1. The main problem statement and justification for the research has not been clearly stated.

Reply:Thanks for your very important comments. We have highlighted the main statements and justification again and made the revised manuscript clear. According your comments, we have highlighted the contribution of the manuscript to the empirical literature (see Page 1 on Line 23 to Line 25) in the revised manuscript. We highlighted and pointed out the novelty of the work in the introduction (see Line 75 to Line 79), and also elaborated the novelty of the work (see Line 95 to Line 101). In addition, we explain the reasons why the heterogeneous MoS2 catalysts can enable a more stable and better electrochemical performance than the expensive Pt/C-RuO2 catalysts (see Line 213 to Line 218).

  1. It is not clear the contribution of the manuscript to the empirical literature.

Reply: Thanks for your comment. To our knowledge, by constructing its own heterogeneous surface/interface and edges, the heterogeneous MoS2 catalysts is the first non-bifunctional MoS2 catalyst be to be used in aqueous Zn-Air battery where the OER reactivity of the single MoS2 catalyst exceeds commercial Pt/C-RuO2 catalytic activity. This work paves the way for the research on improving the performance of aqueous Zn-Air battery by constructing its own heterogeneous surface or interface instead of the commonly reported composite of other materials to form a bifunctional catalyst. We have also highlighted the contribution of the manuscript to the empirical literature (see Page 1 on Line 23 to Line 25, Line 96 to Line 101).

  1. The novelty of the work should be boldly explicated. The main novelty in this work must be clearly pointed out in the introduction.

Reply: Thanks for your comment. Compared with the currently reported bifunctional catalysts (Sustainable Energy Fuels, 2018, 2, 39–67) that consist of two typical materials with different catalytic functions such as Mo–N/C@MoS2 (Adv. Funct. Mater. 2017, 27, 1702300) Co9S8@MoS2 (ACS Appl. Mater. Interfaces 2018, 10, 2, 1678–1689), CoS2@MoS2@NiS2 (Journal of Colloid and Interface Science 610 (2022) 653–662) and so on, this work reported an efficient heterogeneous MoS2 catalyst. The catalyst is only composed of MoS2 that can form heterogeneous interfaces and defects-embedded active edge sites only by using a relatively low synthesis temperature of 180 ℃. Such heterogeneous MoS2 catalyst shows a stable and decent oxygen evolution catalytic activity than commercial and expensive Pt/C-RuO2 catalyst. To our knowledge, this is our first MoS2 catalyst without composite of other materials that exceeds commercial Pt/C-RuO2 catalytic activity for assembled aqueous Zn-Air battery. This work paves the way for the research on improving the performance of aqueous Zn-Air battery by constructing its own heterogeneous surface or interface instead of the commonly reported composite of other materials to form a bifunctional catalyst. We have also highlighted the contribution of the manuscript to the empirical literature (see Page 1 on Line 23 to Line 25, Line 75 to Line 79 and Line 95 to Line 101).

  1. The authors should mention on the concept of this work with the progress against the most recent state-of-the-art similar studies.

Reply: Thank you very much for your reminder. According to your suggestion, we compared the electrochemical performance of the reported aqueous Zn-Air batteries using MoS2-based catalysts.   Notably, the recent reported aqueous Zn-Air batteries are made of bifunctional MoS2-based catalysts that consist of MoS2 catalyst and another catalyst with good OER catalytic activity such as N-doped carbon (Ref. 1), Co9S8 (Ref. 2) and CoS2 and NiS2 (Ref. 3). The Zn-Air batteries using the heterogeneous MoS2 catalysts show a long cycle stability (500 cycles, 180 h) and a low discharge voltage (1.88 V) and voltage gap (0.58 V) with a limited capacity of 330 mAh g-1. Even at a limited capacity of 3300 mAh g-1,the  Zn-Air battery using the heterogeneous also shows a acceptable oxygen reduction catalytic and oxygen evolution activity (with 1.01 V voltage gap) than the reported bifunctional MoS2/N-doped carbon,Co9S8/MoS2 and CoS2/MoS2/NiS2 catalysts (with voltage gap from 0.75 V to 0.95 V, see Ref. 1 to Ref 3) with a limited capacity of ~700 mAh g-1. In particular, the hybrid MoS2 catalysts outperform commercial Pt/C in the practically meaningful high-current region (>5 mA cm−2). We also added this Table in the revised Supplementary Material (See Table 1).

Table 1. Comparison of electrochemical properties of Zn-Air batteries using MoS2-based catalysts

[1] Adv. Funct. Mater.2017, 27, 1702300

[2] ACS Appl. Mater. Interfaces 2018, 10, 2, 1678–1689

[3] Journal of Colloid and Interface Science, 2022(610), 653-662,

  1. Figure 2a (STEM) is not very clear to observe. Can the authors replace it by one of better quality?  

Reply: Thanks for the comment. We have characterized the sample using aberration-corrected high-angle annular dark-field scanning TEM (HAADF-STEM) again. However, we still haven't got a clearer STEM image to replace Figure 2a. Notably, the scale unit is 5 nanometers in Figure 2a where we can clearly observe the MoS2 nanosheet structures below 1 nm.

  1. Line 153: Further, the STEM of MoS2 catalyst shows a clear heterogeneous surface consisting of active edge sites and abundant defects (see Figure 2g)…..if I am not mistaken, I see more disordered and the defect highlighted in Figure2 g I am not sure is clearly seen.

Reply: Thanks for the comment. We have modified these descriptions. The reported work shows similar edge and defines these as defects (ACS Nano. 2019, 25, 13(6):6824-6834).

  1. Line 182: “It indicates that the heterogeneous MoS2 catalysts can enable a more stable and better electrochemical performance than the expensive Pt/C-RuO2 catalysts for the assembled aqueous Zn-Air battery.”…why? A reasonable explanation should be given.

Reply: Thanks for your comment. We reply to your questions from the following aspects. MoS2 is first a good electro-catalysts for hydrogen evolution reaction (HER), oxygen evolution reaction and metal-oxygen batteries as the previous reported research work [see Adv. Mater. 2016, 28(10), 1917−1933; Adv. Mater. Interfaces 2016, 3(9), 1500669 and Nanoscale 2018, 10 (47), 22549−22559]. Importantly, the intrinsic catalytic activity of MoS2 is determined by the adsorption ability of the edged Mo sites toward O2 species. The edge sites of the MoS2 nanosheets usually display stronger adsorption toward oxygen (O2) and other intermediates, which leads to a high electro-catalytic activity for the reversible electrochemical conversion [see ACS Nano 2016, 10, 2167−2175; Nanoscale 2018, 10, 22549−22559 and Nano Energy 2019, 65, 103996]. In addition, defect engineering and heteroatom doping are also used to improve the electrochemical activity of MoS2 to enhance the performance of metal-oxygen batteries [see Front. Energy Res. 2020, 8, 109; Nano Energy 2019, 65, 103996 and Chem. Eng. J. 2020, 382, 122854]. Notably, the presence of amorphous phase makes it easier to maintain high electrochemical activity for MoS2 catalyst. For example, Wu et al., have provided both experimental and theoretical evidence for the importance of the short Mo @ Mo bond structures of 1T and amorphous MoS2 in comparison to crystalline MoS2 for explaining the higher electrochemical performance [see the work published on ChemSusChem 2019, 12, 4383-4389]. Whereas crystalline 1T-MoS2 stabilized by intercalated Li+ also displays high performance, Li ions were found to dissolve in the electrolyte during electrochemical testing, resulting in a slow transformation back to the 2H-MoS2 phase and a concomitant decrease in HER activity. In contrast, amorphous MoS2 retains much of its high HER activity during prolonged operation [see ChemSusChem 2019, 12, 4383-4389]. As a result of these advantages mentioned above, the heterogeneous MoS2 can be used as one of the potential candidates for the O2 cathode catalyst with stable and good electrochemical performance, especially for energy storage batteries [Nano Energy 2019, 65, 103996 and Energy & Fuels 2021 35 (7), 5613-5626].

  1. Some part of the text lacks explanation about assertive statements. In addition, some discussion also involving other battery systems should be compared highlighting some pros and cons. The 2 references can help the authors with that. (https://doi.org/10.3390/nano12223933; https://doi.org/10.1021/acsomega.2c06054).

Reply: Thanks for your reminder. According your comments, we explain the reasons why the heterogeneous MoS2 catalysts can enable a more stable and better electrochemical performance than the expensive Pt/C-RuO2 catalysts (see Line 213 to Line 221). In addition, we elaborated the novelty of the work (see Line 95 to Line 101), pointed out the novelty of the work in the introduction (see Line 75 to Line 79), and also highlighted the contribution of the manuscript to the empirical literature (see Page 1 on Line 23 to Line 25). Also, we have added the explanation about assertive statements and also added the two work in the revised manuscript (see Ref [8] and Ref [9]).

  1. Figure 4d is not discussed and it has to do so.

 Reply:Thank you very much for your reminder. We have discussed the Figure 4d in the revised manuscript (see Line 236 to Line 238).

  1. There is not Supporting Information file, or at least I had no access to it.

Reply:Thank you very much for your reminder. We have revised the manuscript and linked the Supporting Information to the corresponding description in the revised manuscript (see Line 82, Line 151, Line 204 and Line 231 marked yellow in the text in the revised manuscript).

Round 2

Reviewer 1 Report

The authors have made the necessary changes and the article can be published.